# Muscle Damage in Dystrophic mdx Mice Is Influenced by the Activity of Ca^2+^-Activated K_Ca_3.1 Channels

**DOI:** 10.3390/life12040538

**Published:** 2022-04-05

**Authors:** Marta Morotti, Stefano Garofalo, Germana Cocozza, Fabrizio Antonangeli, Valeria Bianconi, Chiara Mozzetta, Maria Egle De Stefano, Riccardo Capitani, Heike Wulff, Cristina Limatola, Myriam Catalano, Francesca Grassi

**Affiliations:** 1Department of Physiology and Pharmacology, Sapienza University of Rome, 00185 Rome, Italy; marta.morotti@uniroma1.it (M.M.); stefano.garofalo@uniroma1.it (S.G.); riccardo.capitani@uniroma1.it (R.C.); myriam.catalano@uniroma1.it (M.C.); 2Istituto di Ricovero e Cura a Carattere Scientifico (IRCCS) Neuromed, 86077 Pozzilli, Italy; germana.cocozza@uniroma1.it (G.C.); cristina.limatola@uniroma1.it (C.L.); 3Institute of Molecular Biology and Pathology-National Research Council (CNR), Department of Molecular Medicine, Sapienza University of Rome, 00185 Rome, Italy; fabrizio.antonangeli@uniroma1.it; 4Institute of Molecular Biology and Pathology-National Research Council (CNR), Department of Biology and Biotechnology, Sapienza University of Rome, 00185 Rome, Italy; valeria.bianconi@uniroma1.it (V.B.); chiara.mozzetta@uniroma1.it (C.M.); 5Department of Biology and Biotechnology, Sapienza University of Rome, 00185 Rome, Italy; egle.destefano@uniroma1.it; 6Department of Pharmacology, University of California, Davis, CA 95616, USA; hwulff@ucdavis.edu; 7Laboratory Affiliated to Istituto Pasteur Italia, Department of Physiology and Pharmacology, Sapienza University of Rome, 00185 Rome, Italy

**Keywords:** Duchenne muscular dystrophy, macrophages, fibroblasts, fibrosis, Kcnn4, K_Ca_3.1, grip strength, hanging time, neuromuscular junction, fiber size

## Abstract

Duchenne muscular dystrophy (DMD) is an X-linked disease, caused by a mutant dystrophin gene, leading to muscle membrane instability, followed by muscle inflammation, infiltration of pro-inflammatory macrophages and fibrosis. The calcium-activated potassium channel type 3.1 (K_Ca_3.1) plays key roles in controlling both macrophage phenotype and fibroblast proliferation, two critical contributors to muscle damage. In this work, we demonstrate that pharmacological blockade of the channel in the *mdx* mouse model during the early degenerative phase favors the acquisition of an anti-inflammatory phenotype by tissue macrophages and reduces collagen deposition in muscles, with a concomitant reduction of muscle damage. As already observed with other treatments, no improvement in muscle performance was observed in vivo. In conclusion, this work supports the idea that K_Ca_3.1 channels play a contributing role in controlling damage-causing cells in DMD. A more complete understanding of their function could lead to the identification of novel therapeutic approaches.

## 1. Introduction

Duchenne Muscular Dystrophy (DMD) is a fatal, X-linked disease, due to nonsense mutations in the dystrophin gene. Analogous mutations have been detected in *mdx* mice (the most widely used animal model of DMD) and in dogs [1]. Due to the absence of dystrophin, muscle sarcolemma is weakened [2], and becomes vulnerable to normally non-noxious factors, including muscle contraction. Thus, in dystrophic muscle there is a continuous activation of the repair responses, normally aimed at muscle healing, resulting in chronic inflammation and fibrosis. In *mdx* mice, disease progression is well characterized: Between 3 and 6–8 weeks of age, massive necrosis takes place, followed by a stable regenerative phase, when muscle deterioration is slow. Steady progression is only observed after the age of 12 months, except for the diaphragm, where cell degeneration is continuous [1,3,4]. Thus, the diaphragm, already in the first weeks of *mdx* mouse life, recapitulates fairly well several features of the human disease.

Resident macrophages are rare in healthy muscles, but circulating monocytes are rapidly recruited upon injury, and differentiate to macrophages in order to promote muscle repair [5,6], through a well-defined sequence of events [7,8]. Initially, pro-inflammatory phagocytic macrophages release large amounts of nitric oxide (NO) at the site of injury and clear debris of dead cells. At this stage, they colocalize with areas of active proliferation of myogenic precursors [5]. During this time, macrophages release a plethora of specific molecules that influence proliferation and differentiation of myogenic precursor cells [9]. Later, pro-healing (or anti-inflammatory/alternatively activated) macrophages appear and accumulate at sites of differentiation and fusion of myogenic precursors [9], reducing the level of muscle-damaging NO and promoting muscle regeneration by favoring proliferation of satellite cells (SCs) and maintenance of the stem population through the production of Klotho and IGF-1 (reviewed by [8]). Moreover, anti-inflammatory macrophages promote differentiation and fusion of human satellite cells into myotubes, both in vitro and in vivo [9]. During muscle repair, extracellular matrix deposition by fibroblasts transiently increases, to promote reparative processes [10].

In dystrophic muscles, in which cycles of necrosis and repair take place continuously, macrophages are constitutively present [8] and iNOS expression is detected during the early acute phase [11,12]. Thus, macrophages are likely to contribute to progression of muscle damage in *mdx* mice [12] and possibly also in DMD patients [8]. Accordingly, corticosteroids, which dampen the inflammatory immune response, are the recommended standard of care for DMD treatment [13].

In the context of DMD, the main drawback in pushing macrophages towards a pro-healing phenotype might be that they release TGF-β1 and inhibit apoptosis of activated Fibro-Adipogenic Precursor cells (FAPs), which are free to expand and differentiate into fibroblasts, causing muscle fibrosis [8]. In DMD patients, abundance of alternatively-activated macrophages correlates with enhanced fibrosis and poor prognosis [14]. However, some lines of evidence suggest that, in dystrophic muscle, proinflammatory macrophages release TGF-β1, so that skewing their phenotype in an anti-inflammatory direction can reduce collagen deposition (see [4] for a review).

The calcium-activated potassium channel K_Ca_3.1 (KCNN4; SK4; Gárdos channel) regulates membrane potential and Ca^2+^ signaling in almost all non-excitable cells, such as epithelia, endothelium, fibroblasts and hematopoietic cells, including activated T and B cells and monocytes/macrophages [15]. In particular, K_Ca_3.1 channels contribute to controlling the switch of macrophages and microglia between pro- and anti-inflammatory phenotypes in different microenvironments: blockade of the channels by the selective blocker 1-[(2-chlorophenyl)diphenyl-methyl]-1*H*-pyrazole (TRAM-34) enhances the fraction of anti-inflammatory macrophages in atherosclerotic plaques [16] and microglia in a mouse model of amyotrophic lateral sclerosis [17,18]. In addition, channel block reduces cytotoxic NO production by macrophages, without affecting their phagocytic activity [19].

K_Ca_3.1 channels are also expressed in fibroblasts in several tissues and contribute to TGF-β1-induced fibroblast proliferation by amplification of Ca^2+^ signaling. Ca^2+^-induced K_Ca_3.1 channel opening causes membrane hyperpolarization, increasing Ca^2+^ influx through TGF-β1-activated pathways [20], which involve TRPV4 channels in cardiac fibroblasts [21]. However, channel blockade or even genetical ablation have no adverse consequences in healthy conditions [15,22]. By contrast, in pathological conditions, K_Ca_3.1 channels control fibrosis in several tissues [22]. For example, the block of K_Ca_3.1 channels is evaluated as a possible treatment of fibrosis in tissues as diverse as cornea [23], kidney [24], vasculature [25] and lung [26]. In the heart, blockade of K_Ca_3.1 channels prevents myocardial fibrosis in models of hypertension [27,28]. However, the role of K_Ca_3.1 channels has not been investigated in muscle fibroblasts, in particular in relation to muscular dystrophy. Of note, K_Ca_3.1 channels play no role in myoblasts fusion and muscle repair processes, which are critically dependent on the function of K_Ca_1.1 channels [29].

Since K_Ca_3.1 channels independently influence macrophage phenotype and fibroblast proliferation, acting on these channels can interrupt the cascade of events that causes muscle damage in muscular dystrophy. In this paper, we show that blockade of K_Ca_3.1 channels reduces muscle fibrosis, pushes macrophages towards a pro-healing phenotype and protects dystrophic muscles from damage. In this framework, investigating the role of K_Ca_3.1 channels might provide an innovative approach to better understand molecular mechanisms of muscle degeneration in DMD. 

## 2. Materials and Methods

### 2.1. Animals and Treatments

C57BL/10ScSn-Dmd<mdx>/J (*mdx*) mice were purchased from Jackson Laboratory and used for experiments or for breeding. Animals were housed in standard cages at a constant temperature (22 ± 1 °C) and relative humidity (50%), with a 12:12 h light:dark cycle (light on 07.00–19.00 h). Food and water were available ad libitum. Male mice were randomly divided into two groups and treated 5 days/week by intraperitoneal injections of 120 mg/kg of TRAM-34 or the same amount of vehicle (50 μL, peanut oil, Sigma-Aldrich, St. Louis, MO, USA, #P2144). TRAM-34 was synthesized as described [30]. We treated mice starting at three different ages: 3 weeks-old, 5 weeks-old and 15 weeks-old. One group of mice was used upon arrival from Jackson Lab, after a week of acclimatization (animals treated from 5 to 9 weeks of age), and one after 10 weeks (treatment from 15 to 19 weeks). Mice treated from 3 to 8 weeks of age were born in our facility over a period of 6 months; each subgroup contained at least one TRAM-34-treated and one vehicle-treated mouse from the same litter. At the end of treatments, mice were overdosed with halothane and then intracardially perfused with phosphate buffered saline (PBS). Experiments were authorized by the Italian Ministry of Health (Authorization n. 320/2020).

### 2.2. Grip Strength and Hanging Test

Grip strength was measured using a meter (Ugo Basile, Gemonio, Italy, #47200) consisting of a grasping grid fitted to a force transducer. Mice were held at the base of the tail and allowed to grab the grid with the forelimbs. Mice were then pulled gently backwards until they released the grip. The peak force achieved in 5 trials was taken as a measure of the grip strength. For the hanging test, mice were allowed to grab a horizontal wire with their front paws and the time spent hanging was measured (maximum time allowed, 600 s; [31]). Behavior was scored according to the following scale: (1) hanging onto the bar with both forepaws; (2) in addition to 1, attempted to climb onto the bar; (3) hanging onto the bar with two forepaws and one or both hind paws; (4) hanging onto the bar with all four paws with tail wrapped around the bar; (5) able to walk on the bar to escape [17,32]. When animals fell off the wire, they were returned to the wire without stopping the timer ([31]), for a maximum of 3 times, to reduce the risk of fall-related injuries. Both tests were performed once a week, starting from 3 weeks of age, to avoid habituation.

### 2.3. Immunofluorescence

Diaphragm and tibialis anterior muscles were isolated, fixed in 4% paraformaldehyde and snap frozen in isopentane at −80 °C. Cryostat sections (10 μm) were washed in PBS with Ca^2+^ and Mg^2+^, blocked (3% goat serum in 0.3% Triton X-100) for 1 h, at RT, and incubated overnight at 4 °C with specific antibodies diluted in PBS containing 1% goat serum and 0.1% Triton X-100. The sections were incubated with the following primary Abs: Iba1 (Wako, Osaka, Japan, #019-19741, 1:500), Arg1 (Santa Cruz biotechnology, Santa Cruz, CA, USA, #sc-271430, 1:500), Collagen 3A1 (Santa Cruz biotechnology, #sc-271249, 1:500), Laminin (Sigma Aldrich, St. Louis, MO, USA, #S-L9393, 1:500). After several washes, sections were stained with the fluorophore-conjugated antibody and Hoechst for nuclei visualization. For Iba1/Arg1+ staining, sections were first boiled for 20 min in citrate buffer (pH 6.0) at 95–100 °C. Images were acquired using a CoolSNAP camera (Photometrics, Tucson, AZ, USA) coupled to an ECLIPSE Ti-S microscope (Nikon, Tokyo, Japan) and processed using MetaMorph 7.6.5.0 image analysis software (Molecular Device, San Jose, CA, USA), after background subtraction. Images were scored in a single-blinded manner. Signal co-localization of Iba1^+^ and Arg1^+^ was analyzed measuring the average fluorescence intensity (pixel) of merged signals. Collagen 3A1 was quantified as the ratio of the area occupied by fluorescent cells to the total slice area.

To quantify fiber size, entire muscle cross sections were imaged with 10^x^ magnification and scanned. Images were acquired and aligned using MetaMorph 7.6.5.0 and analyzed using Myosoft plugin for ImageJ [33]. We measured minimal Feret’s diameter (the minimum distance between parallel tangents at opposing borders of the muscle fiber), an indicator little influenced by small variations in sectioning angle [34]. The number of fibers with centrally located nuclei was counted manually in the central part of the slice.

### 2.4. Neuromuscular Junction Evaluation

To evaluate the fragmentation of NMJ, fixed tibialis anterior muscles were sectioned longitudinally (20 μm) and incubated with rhodamine-conjugated alpha-bungarotoxin (α-BTX; Molecular Probes, Thermo Fisher Scientific, Waltham, MA, USA, B13423; 6 mg/mL) in DMEM plus 20% FBS for 50 min at 37 °C. Slices were extensively washed with the same medium, then with PBS and imaged as above. Images of acetylcholine receptors (AChRs) stained by α-BTX were used for the quantification of NMJ fragmentation. Junctions were defined as continuous if AChRs formed a continuous ribbon with 0–2 interruptions, or fragmented if 3 or more AChR clusters were present. Images were acquired using MetaMorph 7.6.5.0 and analyzed using a deconvolution package.

### 2.5. PCR

Total RNA was extracted from cultured C2C12 cells (see below for details) and fibroblasts, from purified SCs and primary murine microglia and from diaphragm muscles of vehicle and TRAM-34-treated *mdx* mice following a standard Trizol Reagent (Invitrogen, Thermo Fisher Scientific, #T9424). DNA contamination was removed with DNase I (Invitrogen), according to the manufacturer protocol. Quantification was performed with NanodropOne (Thermo Fisher Scientific) and 1 µg of total RNA was reverse-transcribed using IScript Reverse Transcription Supermix (Bio-Rad, Hercules, CA, USA #1708841). The reverse transcription product was used as a template for PCR amplification. PCR was carried out in a MJ-Mini-Personal Thermal Cycler (Bio-Rad) using Dream Taq Green PCR Master Mix (Thermo Fisher Scientific). All reagents were added to reaction tubes according to manufacturer’s suggestions; primers were added at a final concentration of 0.5 µM.

For the reverse transcript PCR, primer sequences were: gapdh, forward: 5′-TCGTCCCGTAGACAAAATGG-3′, reverse: TTGAGGTCAATGAAGGGG; kcnn4, forward: GGCTGAAACACCGGAAGCTC, reverse: CAGCTCTGTCAGGGCATCCA. The PCR reaction was as follow: 95 °C for 1 min, 40 cycles of 95 °C for 15 s, 60 °C for 30 s, 72 °C for 1 min, followed by a final extension step at 72 °C for 5 min. Amplification product was analyzed on 2% agarose gel, molecular weight was calculated with reference to a 1000 bp molecular marker (Bio-Rad #170-8202).

For the Real Time PCR (RT-PCR) the primer sequences are listed in Appendix A. The reaction was carried out in a I-Cycler IQ Multicolor RT-PCR Detection System (Bio-Rad, #172-5201) using SsoFast EvaGreen Supermix (Bio-Rad). Each sample was assayed in duplicate. Relative gene expression was calculated by ΔΔCT analysis relative to GAPDH expression levels.

### 2.6. Isolation of Muscle Satellite Cells

Mononucleated cells were obtained from hindlimb muscles of *mdx* mice aged 3 or 8 weeks, sacrificed by cervical dislocation. The muscles were collected then washed in Ca^2+^- and Mg^2+^-free PBS, minced with scissors, and dissociated enzymatically with type I collagenase (Sigma Aldrich, St. Louis, MO, USA, 2 mg/mL) plus dispase type II (Roche, Basel, Switzerland, 100 mg/mL) in MEM (Gibco, Thermo Fisher Scientific, #11095-080) for 90–120 min in a shaking bath. Enzymes were inactivated adding Hanks’ Balanced Salt Solution (Gibco). Cells were filtered using nylon strainers with decreasing size (100 μm, 70 μm, 40 μm; Falcon, Thermo Fisher Scientific), centrifuged at 300× *g* for 5 min at 4 °C and counted. Satellite cells were then sorted using satellite Cell Isolation Kit (Milteny Biotec, Bergisch Gladbach, Germany), according to manufacturer’s instructions.

### 2.7. Flow Cytometric Analysis

Fluorochrome-conjugated mAbs raised against the following antigens (clone name indicated in parentheses) were used for the flow cytometric analysis of single cell suspensions from muscle: CD45-APC-eFluor 780 (30-F11), CD11b-FITC (M1/70), F4/80-PerCP-Cyanine5.5 (BM8), MHC Class II (I-A/I-E)-APC (M5/114.15.2), CD206/MMR-PE (MR6F3) from eBioscience^TM^-Invitrogen. Cells were washed and suspended in staining buffer (PBS, 0.5% BSA, 2 mM EDTA, 0.025% NaN_3_). Zombie Violet™ Fixable Viability Dye from BioLegend (San Diego, CA, USA) was used to exclude dead cells and anti-CD16/32 (clone 2.4G2) was added (10 min) to prevent nonspecific and Fc-mediated binding. Cells were stained with the indicated antibodies for 20 min at 4 °C. CD206 was stained intracellularly by using the eBioscience™ FoxP3/Transcription Factor Staining Buffer Set according to the manufacturer’s instructions. Samples were analyzed by FACS-CantoII (BD Biosciences, Milano, Italia), and data elaborated using FlowJo software v.10.7.1 (FlowJo, Ashland, OR, USA). Gating strategy to identify macrophages is shown in Appendix A.

### 2.8. Primary Fibroblast Culture and Proliferation Assay

Fibroblast cultures were obtained from hindlimb muscles of *mdx* mice at postnatal day 10 (P10) following established procedures [35]. Muscles were minced with scissors, digested with type I collagenase (2 mg/mL, Sigma) for 45 min at 37 °C in MEM (Gibco, #11095-080), then equilibrated in S-MEM (Gibco, #11380-037) Ca^2+^ -free dissociation medium with added HEPES (20 mM, Sigma) for 15 min at room temperature and mechanically dissociated by repetitive aspiration through a Pasteur pipette. Fibroblasts, which adhere to the support more rapidly than myoblasts, were obtained by pre-plating the cell suspension into 100 mm Petri dishes (Corning, Corning NY, USA). After 1 h, the supernatant was discarded and replaced by fresh Dulbecco’s modified Eagle’s medium (DMEM, Gibco) plus 10% fetal bovine serum (FBS, Sigma, St. Louis, MO, USA) and 2% pen/strep for long-term cultures. For proliferation assays, fibroblasts were detached and seeded into 35 mm well plates (2.5 × 10^4^ cells/well) and grown until 60% confluent. Cells were then starved for 24 h in serum-free medium, then returned to medium containing 10% FBS and stimulated with TGF-β1 (Peprotech, Cranbury, NJ, USA, #100-21; 10 ng/mL) in the presence of TRAM-34 (2.5 μM) or vehicle (DMSO, 0.25 μL/mL). After 0 h, 24 h, 48 h, 72 h cells were mobilized with 0.1% trypsin and counted using a standard hemocytometer. All experiments were performed in duplicate.

### 2.9. Proliferation and Fusion of C2C12 Cells

Cells of the murine-derived muscle cell line C2C12 (obtained from ATCC, American Type Culture Collection) were cultured in a growth medium, composed of DMEM supplemented with 20% FBS and 2% Pen/Strep. For proliferation assay, 2 × 10^4^ C2C12 cells were seeded into 35 mm well plates in growth medium and allowed to proliferate in the presence of either TRAM-34 (2.5 μM in DMSO) or DMSO alone (0.25 μL/mL). After 0 h, 24 h, 48 h, 72 h cells were mobilized with 0.1% trypsin and counted using a standard hemocytometer. All experiments were performed in triplicate. To determine fusion index, C2C12 cells (in 35 mm Petri dishes) were allowed to grow up to 80–90% confluency, then induced to differentiate by serum deprivation and treated with TRAM-34 (2.5 μM) or vehicle. After 48 h cells were fixed with 4% paraformaldehyde and stained with cell-permeant Hoechst 33,342 dye for nuclei visualization.

### 2.10. Statistics

Data are presented as mean ± standard deviation. Statistical significance was analyzed using the non-parametric Mann–Whitney U-test. Effect size has been estimated by Cohen’s *d* formula [36]:$$d=\frac{\left|m_{C}-m_{T}\right|}{\sqrt{\frac{\left(n_{C}-1\right)\;SD_{C}^{2}+\left(n_{T}-1\right)SD_{T}^{2}}{n_{C}+n_{T}-2}}}$$
where *m_C_*, *m_T_*, *SD_C_*, *SD_T_* are mean and standard deviation of control and treated groups, comprising *n_C_* and *n_T_* mice, respectively. For most experiments reported here, *n_C_* = *n_T_* = 10, so we used a pooled SD value to calculate *d* value, which is likely to provide a better estimate of the population standard deviation [37].

## 3. Results

### 3.1. K_Ca_3.1 Channels Affect Macrophage Phenotype in mdx Muscles

In this work, we took advantage of the well understood pathology development in *mdx* mice, to test our working hypothesis. To address the phase of active necrosis, which most closely resembles human disease, we treated *mdx* mice with TRAM-34 starting as soon as animal weight allowed weaning at about 3 weeks of age (8.7 ± 1.2 g, *n* = 27; age between 20 and 25 days). Other mice were treated at later ages (5 to 9 weeks and 15 to 19 weeks), to define which effects were dependent on the disease stage (Figure 1). Unless otherwise specified, data refer to the youngest animals.

Given the involvement of K_Ca_3.1 channels in modulating the phenotype of macrophages, we investigated the effect of K_Ca_3.1 channel block in *mdx* mice considering the expression of CD206 (the macrophage mannose receptor) as a major marker of reparative M2-polarized cells. The cytofluorimetric analysis of macrophages from the hindlimb muscles of *mdx* mice highlighted a significant increase of CD206 expression after TRAM-34 treatment (Figure 2A), suggesting a possible reparative activity of macrophages promoted by K_Ca_3.1 channel blockade.

A further indication of a shift towards an anti-inflammatory milieu upon blockade of K_Ca_3.1 channels was obtained by RT-PCR analysis of the diaphragms isolated from vehicle- and TRAM-34-treated mice. In particular, the expression of the anti-inflammatory genes arginase 1 (*Arg1*) and *CD206* were increased, while the expression of *iNOS*, a pro-inflammatory gene, decreased by about 30% (Figure 2B).

For a more detailed analysis, we examined macrophage infiltration of the diaphragm by Arg1 and Iba1 immunofluorescence. Treatment with TRAM-34 induced a significant increment of Arg1^+^ cells among macrophages (quantified as the ratio of Arg1^+^/Iba1^+^ cells), with an increase in the density of macrophages (Iba1^+^ cells/ mm^2^; Figure 2C and Appendix A). The enhancement of Arg1 expression is a hallmark of the increased metabolism of arginine, which typically mirrors the anti-inflammatory macrophage phenotype [38,39]. These results thus support our working hypothesis.

To define whether the influence of K_Ca_3.1 channels on macrophage phenotype is specific for the active phase of pathology, two groups of mice were treated at later ages (Figure 1). Also in these animals, there was a significant increase in Arg1^+^/Iba1^+^ cells infiltrating the diaphragms of TRAM-34 treated vs. vehicle-treated control animals (Figure 2C). This occurred in spite of a low ratio of Arg1^+^/Iba1^+^ cells in the 5–9 weeks group as compared with the others, indicating that K_Ca_3.1 channels contribute to influencing macrophage phenotype independently of the disease stage.

### 3.2. K_Ca_3.1 Channels Influence Fibrosis in mdx Muscles

By skewing macrophage towards an anti-inflammatory phenotype, the blockade of K_Ca_3.1 channels possibly favors fibrosis, which is a key step in the reparative process following acute injury [8]. However, blockade of K_Ca_3.1 channels has a notable anti-fibrotic effect [22], that might overcome the possible pro-fibrotic effect of macrophages. We therefore examined the effect of the treatment on collagen deposition in the same *mdx* animals used for macrophage characterization. The collagen-covered surface in transverse sections of the diaphragm was reduced by about 30% in animals treated with TRAM-34, as compared to vehicle-treated animals (Figure 3A and Appendix A). Analogous results were obtained also in the tibialis anterior muscle (Figure 3A and Appendix A) and reproduced in both muscles when animals were treated at later ages (Figure 3A; Appendix A), confirming that the blockade of K_Ca_3.1 channels opposes muscle fibrosis. In line with this reduced deposition of extracellular matrix, gene expression of collagen and fibronectin was reduced in TRAM-34 vs. vehicle-treated mice (Figure 3B). Comparing the two groups of animals closer in age (3–8 weeks vs. 5–9 weeks), the percentage of collagen-covered surface was significantly different in the Tibialis anterior (*p* = 0.01), but not in the diaphragm (*p* = 0.057) (Figure 3A). However, the difference between control animals had no influence on the effects of K_Ca_3.1 channel block.

K_Ca_3.1 channels are expressed in fibroblasts, with a proliferation-promoting function in several tissues, but this role has not been confirmed in fibroblasts derived from *mdx* muscle, which in culture proliferate less than wt fibroblasts [35]. We therefore performed experiments on cultures of *mdx* muscle fibroblast. PCR experiments showed that the kcnn4 gene is expressed in these cells (Appendix A). After a single proliferative stimulus (TGF-β1 10 ng/mL on day 0), cells treated daily with TRAM-34 (2.5 μM) proliferated less than vehicle-treated cells (Figure 3C), in agreement with the accepted role of K_Ca_3.1 channels in fibroblasts. By contrast, channel blockade by TRAM-34 had no effect on the proliferation and differentiation of myogenic C2C12 cell line over 3 days (Appendix A). Indeed, kcnn4 expression was detected neither in C2C12 cells nor in satellite cells isolated from *mdx* muscles (Appendix A).

Taken together, these data indicate that K_Ca_3.1 blockade by TRAM-34 reduces both fibroblast proliferation in response to TGF-β1 and collagen deposition in the extracellular matrix of dystrophic muscle. At the same time, treatment is not expected to have direct negative effects on the function of satellite cells and hence on muscle repair.

### 3.3. Block of K_Ca_3.1 Channels Affects Muscle Morphology, but Has No Effect on Performance

The repeated cycles of muscle degeneration and regeneration, typical of dystrophic muscle, result in the presence of regenerating fibers with centrally located nuclei, which represent a cumulative marker of necrosis/regeneration cycles [40]. An excess of small and hypertrophic fibers and increased dispersion in size are also typical [34,41]. The blockade of K_Ca_3.1 channels impacted all these aspects, although the total number of fibers present in muscle cross sections was unaffected (1333 ± 213 fibers/section in controls, 1368 ± 276 in TRAM-34-treated animals, 10 mice per group).

In *mdx* mice treated with TRAM-34, the percentage of fibers with centrally located nuclei was about 30% less than in vehicle-treated animals, in the diaphragm and in tibialis anterior (Figure 4A). The distribution of fiber sizes was also different in animals receiving TRAM-34 as compared to controls. The distribution of minimal Feret’s diameter became narrower, yielding a reduction of its variance coefficient *z* (Figure 4B and Appendix A). No consistent effect on these parameters was observed when treatment was started towards the end of the necrotic phase (5–9 weeks; Appendix A) or during the regenerative phase (15–19 weeks; Appendix A).

Degeneration of muscle fibers has been associated also with damage to the neuromuscular junction (NMJ), which becomes fragmented in dystrophic muscles [42]. We therefore analyzed the appearance of the NMJ in longitudinal sections of tibialis anterior. Only about 20% of the examined NMJs had a normal appearance, in both vehicle- and TRAM-34-treated mice; all the others were discontinuous and more than half showed more than 4 fragments (Figure 4C).

We also tested the outcome of channel blockade on muscle performance in *mdx* animals, measuring forelimb grip strength and the ability to hang to a suspended wire (hanging time). There was a large interindividual variability for both parameters, but no statistically significant difference in performance between animals receiving TRAM-34 or vehicle over the duration of the treatment (Figure 5).

At baseline (3 weeks of age), maximal forelimb strength normalized to body weight was 4.0 ± 1.1 gF/g for animals allocated to the control group (*n* = 14), 4.2 ± 1.2 gF/g for animals in the TRAM-34 group (*n* = 13). As expected for very young animals, values at the end of the treatment (8 weeks) were larger for both groups: 6.28 ± 0.66 gF/g for controls (*n* = 9), 6.1 ± 1.1 gF/g for treated animals (*n* = 10). The lack of a consistent effect was observed also for animals treated at later ages (data not shown).

In the hanging test, all animals performed quite well, except for the first session, at 3 weeks of age, when several animals failed to hang to the suspended wire. In the subsequent sessions, all mice were able to remain on the wire, hang on it with four paws and walk on the wire, reaching score 5, but a large inter-individual variability in the time they spent on the wire was evident. TRAM-34-treated animals remained on the wire somewhat less than vehicle-treated mice (Figure 5), mostly because, after reaching the escape position, mice hung to the wire by the tail or hind limbs and then let themselves fall.

## 4. Discussion

In this study, we analyzed the therapeutic potential of simultaneously targeting macrophage phenotype and fibroblast proliferation by inhibiting K_Ca_3.1 in order to reduce muscle damage in *mdx* mice.

The data reported here show that by blocking channel activity, we can indeed bias macrophages towards an anti-inflammatory phenotype and reduce collagen deposition in the diaphragm and in the tibialis anterior of animals at different ages. In parallel, muscle fibers suffer less damage, but only when treatment spans the whole period of acute necrosis. This is reasonable: a less hostile environment protects the muscle from ongoing damage, but it cannot revert previous cell death. The need for an early start of treatment has been reported also with prednisone [43], a standard DMD treatment.

Based on literature data, the reduction of fibrosis was the most expected effect. K_Ca_3.1 channels have been shown to regulate fibrosis in different tissues and organs, including heart [27,28] and vasculature [25]. Of note, monocyte-derived fibrocytes contribute to the development of fibrosis in both pressure-overloaded heart [28] and dystrophic skeletal muscle [10]. Here, we show that skeletal muscle fibrosis is reduced by 30–40% when *mdx* mice undergo blockade of K_Ca_3.1 channels for 4 or 5 weeks. The increased anti-inflammatory phenotype of macrophages can lead to a reduction of fibrosis, as recently reviewed [4]. However, we report that TRAM-34 reduces proliferation of fibroblasts in vitro, which is likely to occur also in vivo, directly contributing to the reduction of collagen deposition. Whether recruitment of fibrocytes is also reduced, as observed for the heart of hypertensive rats [28], remains to be seen. In comparison to the approaches previously used to treat fibrosis in muscular dystrophy, channel blockade appears to be very effective. For instance, halo-fuginone, a blocker of TGF-β-mediated collagen synthesis, prevents collagen deposition in the diaphragm to an extent similar to that reported here, but has no effect on tibialis anterior [44], while we detected a reduction in all muscles examined and at all animal ages. Losartan and suramin, other antagonists of the TGF-β1 pathway, also reduced fibrosis in *mdx* mouse diaphragms [45,46], but after longer treatments than used here. Metformin also reduces collagen deposition by 25–30% in the tibialis anterior muscle [47,48].

Less obvious was the outcome of TRAM-34 treatment on macrophage phenotype. In the case of microglia, the brain-resident counterparts of systemic macrophages, expression and function of K_Ca_3.1 channels depends on the activating insult [49]. Consistent with this notion, blockade of these channels has been shown to push microglia towards an anti-inflammatory phenotype in a mouse model of amyotrophic lateral sclerosis [17,18] and other diseases, but towards a proinflammatory phenotype in glioma-infiltrating microglia/macrophages [50]. A similar variability apparently exists also for macrophages. The expression of K_Ca_3.1 channels is upregulated in human macrophages polarized in vitro towards both M1 and M2 phenotypes in comparison to unstimulated monocytes [16]. In a study on the instability of atherosclerotic plaques [16], channel blockade by TRAM-34 reduced the expression of M1 markers with little effect on M2 markers in vitro, while in vivo it promoted the expression of Arg1 and decreased expression of CD36, considered a M1 marker. Conversely, the density of neither iNOS^+^ nor CD163^+^ macrophages was altered in a mouse model of myocardial infarction upon treatment with TRAM-34 [51]. Possibly, the chronic vs. acute nature of the insults investigated in the two papers contribute to the discrepancy. In the chronically affected dystrophic muscle, we report the upregulation of some typical anti-inflammatory markers, such as Arg1 and CD206 (revealed both in terms of protein and gene expression), together with the decrease of iNOS gene expression.

Our data show that K_Ca_3.1 gene is not expressed in satellite cells purified from *mdx* mouse muscles and in the myogenic C2C12 cell line. Accordingly, treatment with TRAM-34 did not affect proliferation and differentiation of C2C12 cells. It is thus plausible that K_Ca_3.1 channels contribute to the regulation of macrophage phenotype and fibrosis in dystrophic muscle, as they do in other tissues (see references mentioned above), without directly affecting the function of satellite cells. We show that *mdx* mice treated with TRAM-34 starting at 3 weeks, during the massive necrosis phase, show less fibers with centrally located nuclei, typical of regenerating fibers, although the total number of fibers in muscles is the same. There are also fewer small or hypertrophic fibers, normally found in *mdx* mice [34], suggesting that less degeneration occurs, in line with the shift of macrophages towards an anti-inflammatory, protective phenotype. The degeneration–regeneration cycles already occurred when treatment was started at 5 or 15 weeks, so that no protective effect could take place. In spite of the decreased number of regenerating fibers, there was no reduction in the fragmentation of NMJs in TRAM-34-treated animals as compared to control mice. It is still debated whether the damage to the NMJ is due to the absence of dystrophin or to cycles of damage/regeneration of muscle fibers (see [52] vs. [53] for recent contributions on the two opposing views). Our data suggest that the protection offered by the blockade of K_Ca_3.1 channels is not sufficient to prevent fragmentation of the NMJ.

The lack of effect on in vivo performance does not appear to be due to problems with our measurements, as our data on muscle performance are in line with published results. In the period examined, normalized grip strength increases are in the range expected for this parameter ([54]) for all mice. In the hanging test a large inter-individual variability was also previously reported for wild type mice [55]. Rather, it can be considered an expected finding, as many other compounds with beneficial effects on muscle morphology have no or very limited effects on in vivo performance [32]. Even prednisolone, the gold standard of dystrophy treatment, induces a gain in grip strength only after 16 weeks of treatment [56], at an age when *mdx* mice show a marked decline in force. Most likely, our failure to detect an effect on muscle force is due to the fact that young *mdx* mice have minor functional impairments and perform almost as well as wild type animals in several tasks, including those used in the present study [31,32,54,55,56]. The slightly shorter hanging time of TRAM-34-treated mice was largely due to a behavior indicating a voluntary termination of the task, which is typical of strong animals ([31]). This might be interpreted as suggestive of an increased strength of TRAM-34-treated animals. Possibly, continuing treatment to later ages, when force declines in *mdx* mice, would disclose an effect performance.

Given the mild functional impairments in *mdx* mice at the age used in this study, the absence of improvement in muscle strength should not discourage the consideration that further experiments in this and other preclinical models are warranted for several reasons. First, in DMD, macrophage infiltration and fibrosis occur also in heart, and K_Ca_3.1 channels regulate these processes in other heart pathologies [22]. Moreover, the approach here presented has a good translational potential. A blocker of K_Ca_3.1 channels (Senicapoc) is registered for human use [15]; the interrelation between pro-inflammatory macrophages, muscle fibrosis and prognosis is well established in patients [14] and the age of ambulatory loss can be predicted by the haplotype of the LTBP4 gene, encoding latent transforming growth factor-β binding protein 4 [57], which controls TGF-β1 production by macrophages [47]. Last but not least, treatments alternative to corticosteroids can be useful, given the many adverse effects of these drugs (see references in [43]). Thus, this work supports the idea that the blockade of K_Ca_3.1 channels, alone or in combination with other treatments, can be exploited to reduce inflammation, fibrosis and muscle damage in muscular dystrophy.

In conclusion, we show that K_Ca_3.1 channels represent additional key players in the complex relation among macrophages, fibroblasts and muscle damage in muscular dystrophy. Adequate analysis of their role can deepen the understanding of key pathogenic interactions.

## Figures and Tables

**Figure 1 life-12-00538-f001:**
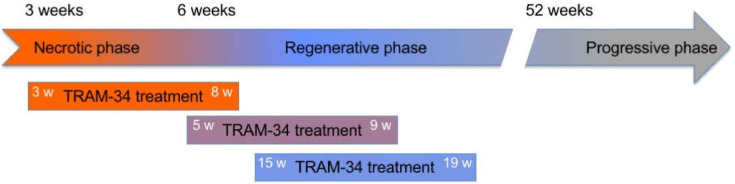
Administration of TRAM-34 in relation to disease stage in *mdx* mice. Each group of mice was treated 5 days a week between the ages indicated (w: weeks): from 3 to 8 weeks of age, from 5 to 9 weeks of age and from 15 to 19 weeks of age.

**Figure 2 life-12-00538-f002:**
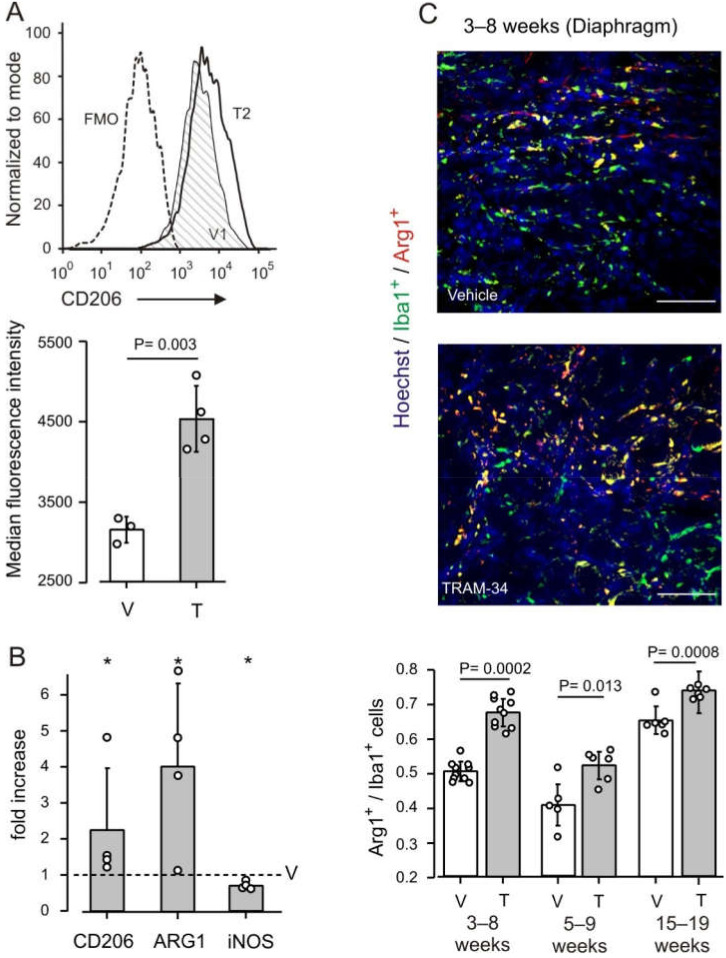
KCa3.1 channels regulate macrophage phenotype (**A**) Modulation of CD206 expression by infiltrating macrophages, identified as CD45^+^CD11b^+^MHCII^+^F4/80^+^ mononucleated cells from hindlimb muscles of 8 weeks old mice. Top panel: representative example of CD206 flow cytometric profile. FMO = fluorescence minus one as negative control; V1 = vehicle-treated animal 1; T2 = TRAM-34-treated animal 2. Bottom panel: median fluorescence intensity of CD206 expression from 3 vehicle-treated (V) and 4 TRAM-34 treated (T) animals. Dots represent individual animals; bars represent the mean value ± SD. P obtained by Student’s *t*-test. (**B**) Relative expression level of Arg1, CD2016 and iNOS genes in the diaphragm of TRAM-34-treated *mdx* mice (age, 8 weeks), compared to each gene in vehicle-treated mice (dotted line). Bars: mean ± SD, dots, individual values in 4 experimental groups. A total of 6 vehicle-treated and 7 TRAM-34-treated animals was examined. *: significantly different from 1 (*p* = 0.016). (**C**) The fraction of arginase-expressing cells is increased in the diaphragm of *mdx* mice treated with TRAM-34 (T) as compared to vehicle (V), independent of animal age. Top panels: representative images. Note the increase in the density of regions Iba1^+^–Arg1^+^ (in yellow); scale bars: 50 μm. Bottom panel: mean ± SD value (bars), and individual data (white circles). N = 10 (3–8 weeks), 6 (6–9 and 15–19 weeks).

**Figure 3 life-12-00538-f003:**
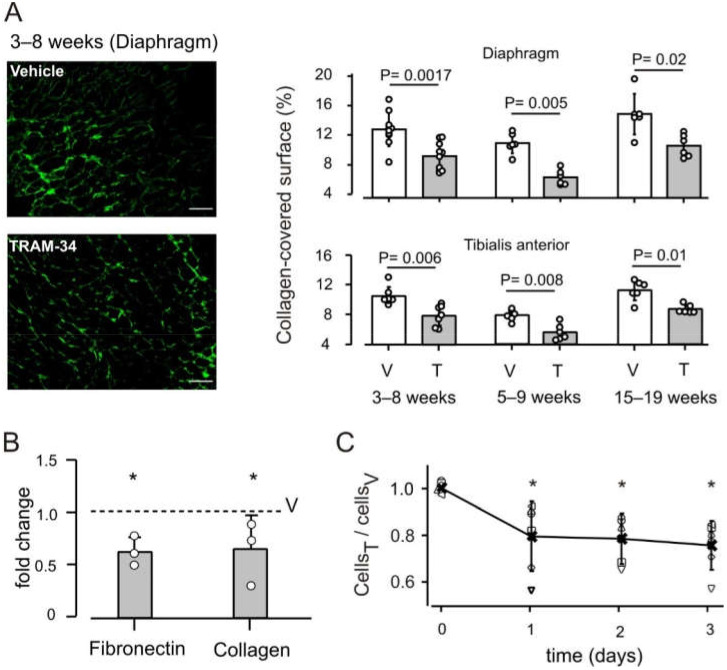
KCa3.1 channels regulate collagen deposition and fibroblast proliferation. (**A**) Collagen-covered surface is reduced in the diaphragm and tibialis anterior of *mdx* mice treated with TRAM-34 (T) as compared to vehicle-treated animals (V). Left panel: representative images; scale bars, 0.1 mm. Right panel: Bars represent the mean values ± SD, small circles represent individual data; treatment at ages as indicated. (**B**) Relative expression level of collagen-1A and fibronectin in the diaphragm of TRAM-34-treated *mdx* mice, compared to each gene in vehicle-treated mice (dotted line). Bars: mean ± SD, dots, individual values in 3 groups of 8 week-old animals. A total of 6 vehicle-treated and 6 TRAM-34-treated animals was examined. *: significantly different from 1 (*p* = 0.036). (**C**) Block of K_Ca_3.1 channels by TRAM-34 reduces proliferation of fibroblasts derived from *mdx* muscles. The results of 6 experiments (each performed in duplicate) are shown as the number of cells in TRAM34-treated dishes divided by the number of cells in vehicle-treated dishes. Crosses represent the average values (± SD), open symbols refer to individual experiments. *: significantly different from 1 (*p* ≤ 0.012).

**Figure 4 life-12-00538-f004:**
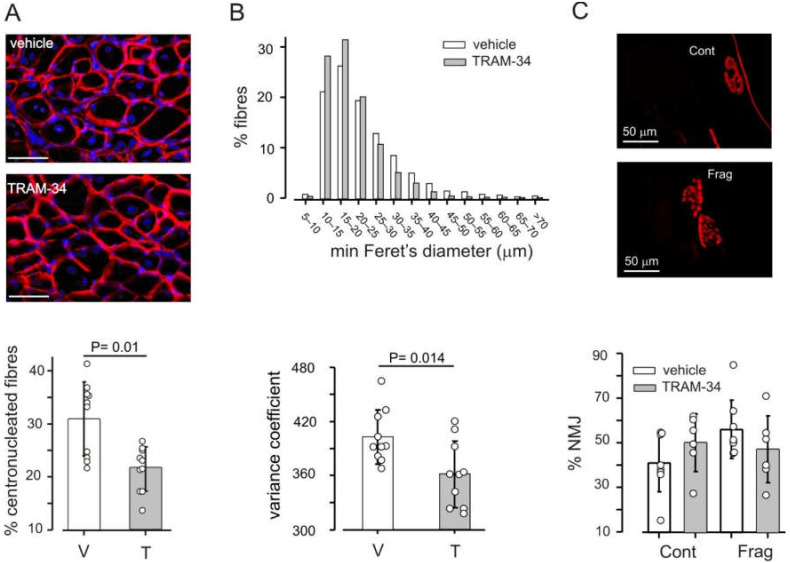
K_Ca_3.1 inhibition influences muscle fibers. (**A**) The percentage of fibers with centrally located nuclei is lower in the diaphragm of *mdx* animals treated with TRAM-34 as compared to controls. Top panels: representative images of sections stained with laminin (red) and Hoechst (blue). Scale bars: 50 μm. (**B**) The distribution (top) of minimal Feret’s diameter is narrower in the diaphragm of *mdx* mice treated with TRAM-34 than vehicle. The variance coefficient is correspondingly reduced (bottom). (**C**) In the tibialis anterior of *mdx* mice, there is the same percentage of continuous (Cont) and fragmented (Frag) NMJs, independent of treatment. Top panels show representative examples of continuous and fragmented NMJs from the same animal. All panels refer to mice treated between 3 and 8 weeks of age. In all bottom panels, bars represent mean values ± SD, dots represent data from individual animals.

**Figure 5 life-12-00538-f005:**
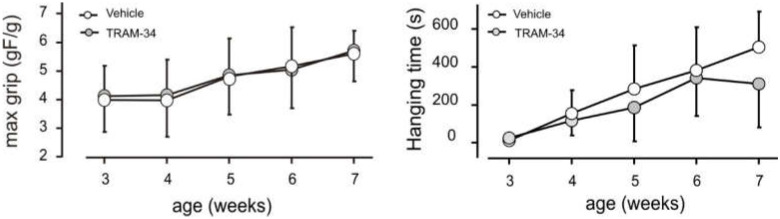
K_Ca_3.1 inhibition has no effect on motor function of *mdx* mice. Analyses of motor function in *mdx* mice treated with TRAM-34 (black symbols) or vehicle (white symbols) with grip strength test (force/weight; **left** panel) and latency to fall in hanging wire test (**right** panel). Data represent mean ± SD. *p* > 0.07 by Mann-Whitney test; *n* = 9 to 14 mice per point.

## Data Availability

Relevant data is contained within the article or Appendix A.

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
