# Peer review of "Muscle Damage in Dystrophic mdx Mice Is Influenced by the Activity of Ca2+-Activated KCa3.1 Channels"

_life, 2022, doi:10.3390/life12040538_

Round 1
Reviewer 1 Report
The paper by Morotti et al describes the attenuation of cellular inflammation and degeneration in the mdx, a Duchenne muscular dystrophy mouse model by blockade of the Kca3.3 ion channel. This study is highly interesting and targets a core problem with Duchenne, the inflammatory response to the dystrophin deficiency, which leads to irreversible fibrofatty replacement of muscle. The study is conceptually well planned and executed and the paper is very well written, a pleasure to read.
Regrettably, the study did not include more or less standardized (for mdx treatment studies) ex vivo electrophysiology experiments of EDL, soleus and perhaps a slice of diaphragm to determine isometric force properties and resilience to eccentric contractions. This would have been an important addition to the grip test. I am mentioning this mostly out of keen interest in this study as it attempts to address a disease progression early on without resorting to attempts to reintroduce dystrophin, which is challenging. For this reason, it would have been nice to see how the treatment fared against other immunomodulating treatments used on the mdx.
I have a few minor comments
Line 114: It is not clear to me for how long the mice were treated. Was a group of 3-8 week old mice treated for a week, or were 3 week old mice treated for 4-5 weeks until the age of 8 weeks. Same question pertains to the two other groups. Obviously, figure 1 can be read in both ways, the legend does no clear this up.
Line 350: “Individual values in 3 groups of animals. A total of 6 vehicle treated….”. If there are untreated and treated, split into three age-groups. Does that mean there is an N=2 in each group. Or should it be read as for each age-group there is a N=6. I suppose the latter considering the number of data points in each bar. Line 350 should be rephrased to clear this up.
Line 390-91: “These results suggest tat Kca3.3 channels contribute to muscle damage…” First of all, this has not been shown and would an oversimplification of a very complex disease mechanism. Blocking the channel improves the phenotype in certain areas, especially by reducing the fiber degeneration and fibrofatty replacement. However, this does not imply that its normal function leads to muscle damage. I would strongly suggest to delete this sentence.
Line 414-15: “This increment…”. Generally, unless explanatory for why something has been done as in methods, explanatory sentences referring to results with a reference belongs to the discussion.
Line 421: same as above. This belongs to the discussion.
Line 424-26: same as above. This belongs to the discussion.
The discussion about the lack of effect on force production exposes a major issue with many treatment studies as well as the problem of when to stop the study and analyze the effect of the treatment. Based on the three treatment groups, I think the authors should elaborate (qualified speculation) on the probable or potential outcome if the treatment was continued for a longer period of time (12 weeks), or treated for the same amount of time and left to grow until 15-16 weeks of age. Can the effect of TRAM-34 be maintained or will the relentless degeneration overcome the benefits?
Author Response
The paper by Morotti et al describes the attenuation of cellular inflammation and degeneration in the mdx, a Duchenne muscular dystrophy mouse model by blockade of the Kca3.3 ion channel. This study is highly interesting and targets a core problem with Duchenne, the inflammatory response to the dystrophin deficiency, which leads to irreversible fibrofatty replacement of muscle. The study is conceptually well planned and executed and the paper is very well written, a pleasure to read.
Regrettably, the study did not include more or less standardized (for mdx treatment studies) ex vivo electrophysiology experiments of EDL, soleus and perhaps a slice of diaphragm to determine isometric force properties and resilience to eccentric contractions. This would have been an important addition to the grip test. I am mentioning this mostly out of keen interest in this study as it attempts to address a disease progression early on without resorting to attempts to reintroduce dystrophin, which is challenging. For this reason, it would have been nice to see how the treatment fared against other immunomodulating treatments used on the mdx.
This is a good point, we have tried to discuss our results in comparison to data present in the literature and we are currently working to extend this study. A sentence in this sense has been added to the discussion (page 13, lines 8-10).
I have a few minor comments
Line 114: It is not clear to me for how long the mice were treated. Was a group of 3-8 week old mice treated for a week, or were 3 week old mice treated for 4-5 weeks until the age of 8 weeks. Same question pertains to the two other groups. Obviously, figure 1 can be read in both ways, the legend does no clear this up.
We are sorry for having been unclear. We better explained the treatment protocol in the text (p. 3, lines 8-11 of paragraph "Animals and treatments") and in the legend of figure 1.
Line 350: “Individual values in 3 groups of animals. A total of 6 vehicle treated….”. If there are untreated and treated, split into three age-groups. Does that mean there is an N=2 in each group. Or should it be read as for each age-group there is a N=6. I suppose the latter considering the number of data points in each bar. Line 350 should be rephrased to clear this up.
We have changed the legend to Figure 3, to clarify that all animals considered were 8 weeks old.
Line 390-91: “These results suggest tat Kca3.3 channels contribute to muscle damage…” First of all, this has not been shown and would an oversimplification of a very complex disease mechanism. Blocking the channel improves the phenotype in certain areas, especially by reducing the fiber degeneration and fibrofatty replacement. However, this does not imply that its normal function leads to muscle damage. I would strongly suggest to delete this sentence.
We thank the reviewer for this observation. We deleted the sentence.
Line 414-15: “This increment…”. Generally, unless explanatory for why something has been done as in methods, explanatory sentences referring to results with a reference belongs to the discussion.
Line 421: same as above. This belongs to the discussion.
Line 424-26: same as above. This belongs to the discussion.
We thank the reviewer for raising these points. We moved the comments to the Discussion (p. 12, lines 8-11 from bottom).
The discussion about the lack of effect on force production exposes a major issue with many treatment studies as well as the problem of when to stop the study and analyze the effect of the treatment. Based on the three treatment groups, I think the authors should elaborate (qualified speculation) on the probable or potential outcome if the treatment was continued for a longer period of time (12 weeks), or treated for the same amount of time and left to grow until 15-16 weeks of age. Can the effect of TRAM-34 be maintained or will the relentless degeneration overcome the benefits?
We thank the reviewer for this point. We added a paragraph on this item to the Discussion (p.13, lines 1-5).
Reviewer 2 Report
This paper uses mdx mice to demonstrate that treatment with a blocker of the KCa3.1 chanels (TRAM-34) can reduce muscle fibrosis, cause macrophages to develop a healing phenotype, and protect against skeletal muscle damage.
The study was well planned and carefully carried out, but the mdx mouse model has limited applicability to human DMD.
General Comments & questions:
- The immune labelling as shown was dificult to see. Could the images be increased so that labelling is more obvious?
- Throughout there are occasions when the past tense is not used to describe observations. This needs correcting.
- Lines 317 & 318 appear to contradict Line 325 & 326. Can you please clarify your statement in lines 317 & 318.
- Fibroblasts are found in tissue other than muscle. Is there evidence that mdx mice have fibroblast pathologies in tissue other than muscle?
- A related question: Do cultured fibroblasts from tissues other than muscle have different properties?
- I understand the phenotype of the mdx model of DMD is different to that observed in humans with DMD. Could this imply the data described in this study might not therefore be relevant to human DMD?
- Was there any evidence that TRAM-34 treatment might have effects on other tissues? Perhaps adverse? Collagen deposition is required in some tissues. eg skin?
- It would be useful to know whether muscle strength WAS improved by treatment in older mdx mice (Lines 503-505), since this is important if TRAM-34 was to be considered as a treatment for children with DMD.
Minor points:
Line 158. Please explain what “Feret’s diameter” is and how it was obtained.
Line 172. Explain what “C2C12” cells are & their source.
Line 173. “Diaphragms” should be singular.
Line 423: Refers to Fig. 5B, but Figure 5 has no “A” or “B” shown.
Author Response
General Comments & questions:
- The immune labelling as shown was dificult to see. Could the images be increased so that labelling is more obvious?
Thank you for pointing out this defect. We have enlarged all micrographs in Figures 2, 3 and 4 to improve data visibility and rearranged Figures 2 and 3 to accommodate the larger panels.
- Throughout there are occasions when the past tense is not used to describe observations. This needs correcting.
We are sorry for the errors, we corrected where necessary.
- Lines 317 & 318 appear to contradict Line 325 & 326. Can you please clarify your statement in lines 317 & 318.
Thank you for noting the apparent contradiction. To clarify this point, we have added a sentence in the Introduction (p. 3, lines 3-5) and in the Results (p. 8, lines 3-4 of paragraph "KCa3.1 channels influence fibrosis in mdx muscle").
- Fibroblasts are found in tissue other than muscle. Is there evidence that mdx mice have fibroblast pathologies in tissue other than muscle?
Actually, fibrosis is secondary to tissue damage due to the lack of dystrophin. Accordingly, fibrosis is prominent in dystrophic skeletal and cardiac muscles. Dystrophin is also expressed in smooth muscle, leading to some measure of gastrointestinal disfunction, but there is one single report of fibrosis in the gut. Furthermore, dystrophin is expressed in brain, retina and Schwann cells, and there are some reports of astrogliosis in the brain, but of course not of fibrosis. A statement about heart involvement has been added to the discussion (p. 13 lines 9-10).
- A related question: Do cultured fibroblasts from tissues other than muscle have different properties?
What we can say in this respect is that cultured mdx muscle fibroblasts proliferate less than wt mouse fibroblast. A similar tendency was noted for human DMD fibroblasts, but the effect was not statistically significant (Reference 35). This is now stated in the text (p. 8, line 9 from bottom).
- I understand the phenotype of the mdx model of DMD is different to that observed in humans with DMD. Could this imply the data described in this study might not therefore be relevant to human DMD?
Of course, the translational potential of studies on mdx mice must be considered with care. Nevertheless, this remains the first pre-clinical model to be used. Since a blocker of KCa3.1 channels is authorized for human use (Senicapoc, see references 15 and 22), it is worth examining the question, although further experiments will be needed. A sentence has been added in the Discussion (p. 13 line 8).
- Was there any evidence that TRAM-34 treatment might have effects on other tissues? Perhaps adverse? Collagen deposition is required in some tissues. eg skin?
Adverse effects of TRAM-34 have never been reported. Apparently, the channel has little role in normal collagen production, but is upregulated in many fibrotic diseases, as detailed in the Introduction (p. 2, lines 2-7 from bottom)
- It would be useful to know whether muscle strength WAS improved by treatment in older mdx mice (Lines 503-505), since this is important if TRAM-34 was to be considered as a treatment for children with DMD.
We are preparing to investigate this point. A sentence on the fact that further experiments are necessary have been added to the discussion (p. 13 line 8).
Minor points:
Line 158. Please explain what “Feret’s diameter” is and how it was obtained.
The details have been added in the Methods section (p. 4, lines 10-11)
Line 172. Explain what “C2C12” cells are & their source.
The details have been added in the Methods section (p. 4 line 1 of paragraph "PCR" and p. 5, line 2 of paragraph "Proliferation and fusion of C2C12 cells").
Line 173. “Diaphragms” should be singular.
Thank you for noting, we have corrected.
Line 423: Refers to Fig. 5B, but Figure 5 has no “A” or “B” shown.
Thank you for noting, we have corrected.